# A Drug Safety Surveillance Study of a Ciprofloxacin/Dexamethasone Ophthalmic Fixed Combination in Peruvian Population

**DOI:** 10.3390/pharmacy9010015

**Published:** 2021-01-10

**Authors:** Homero Contreras-Salinas, Leopoldo Martín Baiza-Durán, Mariana Barajas-Hernández, Alan Omar Vázquez-Álvarez, Lourdes Yolotzin Rodríguez-Herrera

**Affiliations:** 1Pharmacovigilance Department, Laboratorios Sophia, S.A. de C.V, 45010 Zapopan, Jalisco, Mexico; homero.contreras@sophia.com.mx (H.C.-S.); leopoldo.baiza@sophia.com.mx (L.M.B.-D.); mariana.barajas@sophia.com.mx (M.B.-H.); 2School of Medicine and Health Science, Tecnológico de Monterrey, 45138 Guadalajara, Jalisco, Mexico; md.alan.vazalv@gmail.com

**Keywords:** drug safety surveillance, adverse drug reaction, ophthalmic, ciprofloxacin, dexamethasone

## Abstract

(1) Background: drugs provide a significant benefit for patients who require medical treatment; however, their use implies an intrinsic potential danger, with the possibility of causing unwanted effects. These effects are known as adverse drug reactions (ADRs). Post-marketing drug safety surveillance detects unknown risks that have not been identified in clinical trials, and it is necessary to monitor marketed medications under real-life practice. Due to the scarce information about fixed combination of ciprofloxacin 0.3%/dexamethasone 0.1% (SDO), we performed a drug safety surveillance study. (2) Methods: A prospective non-controlled drug safety surveillance study was conducted in Peruvian population. A total of 236 patients prescribed SDO were included derived from 12 sites. Patients’ standardized information was collected through two phone calls, including demographics, medical history, prescribing patterns of SDO, concomitant medication, and ADRs in detail. The ADRs were classified by causality and severity, followed by outcome measures to identify new risk. (3) Results: 236 patients prescribed with SDO participated in the study and 220 were included. A total of 82 ADRs/220 patients were reported after the use of SDO, presenting a ratio 0.37 ADR/patient. The most frequent ADR with SDO administration was eye irritation (30%). All ADRs were classified as non-serious, and 97.5% (*n* = 80) were classified as mild while 2.5% as moderate (*n* = 2). No cases under the severe category were identified. (4) Conclusion: No new risks were found in the population where this study was conducted.

## 1. Introduction

Prescription drugs provide a significant benefit by treating, preventing or diagnosing diseases; however, their use implies an intrinsic potential danger, possibly causing unwanted effects. These effects are known as adverse drug reactions (ADRs) [1,2]. The World Health Organization (WHO) defined ADRs as, “A response to a drug which is noxious and unintended, and which occurs at doses normally used in man for the prophylaxis, diagnosis, or therapy of disease, or the modification of physiological function”, and represent a significant cause of damage that impacts health and economy of patients, their families and society in general [1,3]. ADR monitoring enables the discovery of relevant safety information on an ongoing basis and therefore the identification of both benefits and risks of a specific marketed medication, thus allowing the reduction of the societal burden from adverse drug reactions [4,5] since ADRs can cause a reduction in patient’s quality of life and an important number of deaths annually, with a high economic impact [4,6]. Post-marketing drug safety surveillance is one of the pillars of pharmacovigilance in order to detect unknown risks that have not been identified in clinical trials, since at this stage the drug has been tested for a short period of time on a limited number of individuals. For these reasons, it is necessary to know the behavior of drugs in multiple populations, in uncontrolled environments, in vulnerable groups, during chronic use and when used in combination with other drugs; evaluation which is carried out through the monitoring of marketed medications under real life clinical practice [4].

There are different combinations of antibiotic/steroid ophthalmic medications available for the treatment of ocular infections that have been used for several years [7]. The concomitant use of antibiotics and steroids gained relevance due to the negative effects produced by the severe inflammation caused by ocular infections [8]. The ophthalmic solution containing ciprofloxacin 0.3%/dexamethasone 0.1%, is a combination of a fluoroquinolone, an antibiotic that inhibits bacterial DNA synthesis through its action on topoisomerase II and topoisomerase IV, and a corticosteroid that suppresses inflammation by inhibiting the production of multiple inflammatory cytokines [9]. This ophthalmic fixed combination has demonstrated its aid in controlling post-surgery ocular inflammation, proving to be effective and safe; however, it has been tested on a limited population [10,11].

In this study we characterized the ADRs associated to the use of an ophthalmic fixed combination of ciprofloxacin 0.3%/dexamethasone 0.1% Sophixin DX Ofteno^®^ (SDO) (Laboratorios Sophia, S.A. de C.V., México) in uncontrolled Peruvian population through drug safety surveillance.

## 2. Methods

A prospective non-controlled drug safety surveillance was conducted in Lima, Perú, from February 2019 to April 2020 by 12 collaborating sites. The study’s protocol and its corresponding informed consent form were reviewed and approved by an ethics committee (see Ethics approval section).

Patients were recruited from 27 February 2019 (first enrolled patient) to 29 April 2020 (last patient’s completion). Because this is a non-interventional study, patients who were prescribed SDO by an ophthalmologist (on his/her own initiative) were derived to a member of our team. Afterwards, they were informed about the enrollment process and invited to participate in the study. If the patient agreed to participate, the informed consent was signed.

Over a period of 14 days, a member of our team conducted telephonic interviews with the patients on two different dates, days 7 and 14 after the start of the drug´s administration. On day 7, the personal data were collected (age, gender, nationality, pregnancy or breastfeeding) as well as the characteristics of the drug and its prescription (dose, route of administration, expiration date, batch), data from the patient’s medical history (reason for prescription, concomitant drugs used and their dose, route of administration, start application) and data from any ADR in case they appeared (onset date, description of intensity, ADR duration, need of treatment, re-challenge (if applicable), existence of a similar preceding ADR with the same drug, dechallenge (if applicable), response to dose modification (if applicable), existence of other cause different to drug application that may explain the ADR (if applicable)). During the call that took place on day 14, follow up and general experience with the drug´s use was inquired.

The data collected in each of the scheduled calls was registered in paper and subsequently was recorded an Excel document (Microsoft Office^®^ 365 ProPlus., Washington, Redmond, USA) by Laboratorios Sophia’s Pharmacovigilance Unit and patients were classified according to their age as follows: children (0–12 years old (y/o)), adolescents, (>12–18 y/o), adults (>18–60 y/o), and geriatric (>60 y/o). If any of the patients reported any ADRs, these were classified and evaluated according to severity and causal relationship.

Severity was evaluated in accordance with the Modified Hartwig and Siegel Severity Assessment Scale (Mild, Moderate, Severe) [12,13]. Subsequently, the causal relationship was assessed in accordance with the Naranjo algorithm as: indefinite, probable, possible, doubtful or un-assessable [14]. All ADR were listed in System Organ Class (SOC) and Preferred Term (PT) according to MedDRA v 22.0 (Medical Dictionary for Regulatory Activities).

### 2.1. Outcome Measures

#### 2.1.1. Tolerability

TBD’s tolerability was evaluated by measuring different parameters: patient´s demographics characteristics, prescription and dose, interactions (searched in Micromedex^®^ IBM Corporation 2020, Armonk, NY, USA) [9] safety signals, ADR severity, seriousness, and duration besides ADRs of different genders.

#### 2.1.2. Bibliographic Analysis of ADRs

An analysis was performed comparing the incidence of reported ADRs of this study with those found in two reference drug information databases (Micromedex^®^ IBM Corporation 2020 and MedicinesComplete^®^ “Martindale Drug reference” The Royal Pharmaceutical Society 2020) [9,15]. The frequencies obtained in the databases for individual ophthalmic drugs (ciprofloxacin and dexamethasone) were contrasted with those found in the fixed combination of SDO. On the other hand, a search for interactions with patients who use one or more products concomitantly with SDO were searched in the Micromedex^®^ database.

#### 2.1.3. Statistical Analysis

Quantitative variables were expressed as the mean ± SD, and qualitative variables were described as frequencies and percentages. A binary logistic regression analysis was performed to adjust the variables associated with the dependent variable (ADR) by gender (male and female); comorbidities (“yes” and “no”); prescription (post-surgical, conjunctivitis, chalazion and others); age group (adult and geriatric); and dose (1 drop/4 h, 1 drop/6 h, 1 drop/8 h, 1 drop/12 h, 1 drop/24 h) were used as covariates; moreover, multinomial logistic regression analysis was realized to adjust the variables associated with the dependent variable (ADR) by SDO fixed combination; ciprofloxacin and dexamethasone. All the analyses were made using SPSS version 25 for Mac (IBM, Chicago, IL, USA)

## 3. Results

A total of 263 patients receiving a prescription for SDO signed the informed consent form. Nevertheless, it was not possible to contact 16.3% of the patients (*n* = 43) on account of one of the following: unanswered call (60.5%/*n* = 26) or wrong registered number (39.5%/*n* = 17). A total of 220 patients were contacted; 122 women (children: *n* = 1, adults: *n* = 62, geriatric: *n* = 59), none of them were pregnant; and 98 men (children: *n* = 3, adolescent *n* = 2, adults: *n* = 47, geriatric: *n* = 46) (Table 1).

The results of the study and the databases are shown in Table 2. The most frequent ADR for the three drugs was eye irritation: SDO 30%, dexamethasone 10%, and ciprofloxacin 25% followed by dysgeusia: SDO 1.8% and dexamethasone 10% as well as ocular hyperemia and foreign body sensation in eyes: ciprofloxacin 10% each. Although one patient was found with somnolence in the SDO treatment, no information was found for dexamethasone or ciprofloxacin in the databases.

The 66% (*n* = 145) of patients under treatment with SDO used other simultaneous treatments like artificial tears (*n* = 59; 27%) which were the most commonly used products concomitantly with SDO, followed by ophthalmic NSAIDs (*n* = 20; 9%), antiglaucoma drugs (*n* = 19; 9%), glucose-lowering medications (*n* = 15; 7%), antihypertensive medications (*n* = 13; 6%), monoclonal antibodies (*n* = 7; 3%), and others (*n* = 12; 5%); the remaining 34% of patients were not using any concomitant therapy. With those data a search of different bibliographic sources was conducted, and 21 possible systemic drug interactions with SDO (ciprofloxacin/dexamethasone) were identified: 2 patients used fluoxetine, 7 used oral glucose-lowering medications, and 12 used NSAIDs.

A total of 82 ADRs/220 patients were reported after the use of SDO, presenting a ratio of 0.37 ADR/patient; these ADRs were classified into 3 SOC and 9 PT groups, finding that the most frequent SOC group was eye disorders (93%) and the more frequent PT was eye irritation (75%). The most frequent causality was probable with 72% of cases, followed by possible with 15% (Table 3). The ADRs were classified according to severity as follows: 97.5% as mild (*n* = 80) and 2.5% as moderate (*n* = 2). Moreover, 67% of the patients who presented any ADRs improved in one min or less after the application of the product. No cases under the severe category were identified, in the same way no serious ADRs were identified.

A multivariate analysis was conducted to evaluate the drug’s tolerability in patients with different demographic factors such as, age groups, comorbidities, gender; as prescriptions and the dose of the medication. The results showed a statistically significant increase in the incidence of ADRs in females compared to males (adjusted OR = 2.523, *p* = 0.008, 95% CI 1.28–4.974); however, some factors show a tendency to be associated with the increase in the incidence of ADRs. None of them were statistically significant (Table 4).

With the data in Table 4, a bibliographic comparison between SDO against its individual active ingredients was performed, showing that eye irritation, presenting a statistically significant difference only for dexamethasone (adjusted OR = 0.346, *p* ≤ 0.001, 95% CI 0.202–0.592); but not for ciprofloxacin (adjusted OR = 0.99, *p* = 0.964, 95% CI 0.641–1.528). Likewise, eyelid edema (adjusted OR = 2.376, *p* = 0.482, 95% CI 0.213–26.535), eye pain (adjusted OR = 1.037, *p* = 0.971, 95% CI 0.114–7.47), Ocular hyperemia (adjusted OR = 3.458, *p* = 0.064, 95% CI 0.031–12.839), and Vision blurred (adjusted OR = 1.888, *p* = 0.835, 95% CI 0.235–5.998) were found not statistically significant when comparing SDO with ciprofloxacin or dexamethasone. The incidence of the rest of reported ADRs was below expected (Table 5).

## 4. Discussion

Several sources mention that the fixed antibiotic/steroid combination has been shown to be effective in different infectious, allergic, and inflammatory pathologies [7,16,17]; however, there is limited information about the safety profile of the ophthalmic ciprofloxacin/dexamethasone combination. Nevertheless, we found that SDO is well tolerated by the patients, no patients presenting ADRs discontinued treatment, since the vast majority of these were mild, no serious and receded within a min or less after instillation. Further, in the multivariate analysis no statistically significant differences were found, neither related to demographics, prescription nor to dose. This indicates that there was no increased risk related to its use in the different prescriptions, comorbidities, age groups, as well as to the different dosing schemes administered by the patients in the study Interestingly, we found a statistically significant increase among the patients’ gender, finding that females showed a higher (statistically significant) ADR incidence (0.47 ADR/patient) as compared to males (0.26 ADR/patient). These results coincided with several sources which mention that females present ADRs more frequently than males since differences in multiple factors like pharmacokinetics and pharmacodynamics, adipose tissue, gastrointestinal motility, enzymatic activities [18,19,20,21] could affect the incidence of ADRs in males and females; nevertheless, this incidence had not been reported before for ophthalmic medications.

Although there were 21 possible interactions none of the patients reported any symptoms, this could be explained since doses used in ophthalmic formulations are usually lower than those necessary to cause systemic effects. In addition, even though the presentation of systemic effects following topical instillation of ciprofloxacin and dexamethasone has been reported, this is but a rare instance [22,23,24,25].

A bibliographic analysis comparing SDO’s ADRs to the expected ADRs of each of the individual active ingredients was performed [9,15]. The results showed that eye irritation, which was the most frequent ADR in patients using SDO, was statistically significantly higher than that reported with the use of dexamethasone individually; however, no significant differences were found in patients with ciprofloxacin individually for this ADR. Therefore, for SDO, being a fixed combination of both (ciprofloxacin/dexamethasone), this ADR could be attributable to ciprofloxacin. Additionally, the remaining reported ADRs are either similar to or lower than expected if active ingredients were used separately (Table 5), showing that SDO does not increase the incidence of ADRs compared to individual use.

The identification of safety signals is an important part of the benefit-risk assessment of drugs, for this reason an analysis of the ADRs collected on this study was performed. One unexpected ADR, according to the available published references of the active ingredients of SDO [9,26] was found: a SOC of Nervous system disorders and a PT of Somnolence with doubtful causality. Using the Bradford-Hill criteria (Strength of Association, Consistency, Specificity, Temporality, Biological Gradient Dose-Response, Plausibility, Coherence, Experiment, and Analogy) there is not enough information to support that the application of SDO caused the ADR [27].

Our study’s limitations were that the follow-up carried out via telephone restricts the identification of ADRs that require an ophthalmologist’s expertise; this can cause that most of the reactions were symptoms, and the detection of mild reactions may be limited. Thus, the data collection method through a direct interview to the patient could be limited by the patient’s medical knowledge (for example, characteristics of the prescription and data from the patient’s medical history). On the other hand, because the telephone interviews were carried out at two given times, the patients may not report all the ADRs, considering that their answers could be subject to what the patient remembered at the time of the call. This could be mainly a point to consider in the case of children and older people. However, it has some advantages, such as individualized pharmacovigilance, continuous monitoring and more detailed information on adverse events from a large number of patients.

## 5. Conclusions

In our study, we found no increase in the incidence of ADRs related to SDO use compared to those reported in the literature for its active ingredients administered individually; also, no new risks or safety signals were observed in the population where this study was conducted. Consequentially, a good tolerability safety profile was confirmed.

We identified an increase of ADR in females exposed to systemic drugs supported by the literature but limited, on this regard was available for ophthalmic drugs specifically; nevertheless, more studies are needed to assess these results.

## Figures and Tables

**Table 1 pharmacy-09-00015-t001:** Demographics and clinical characteristics.

		Children	Adolescent	Adult	Geriatric
n		4	2	109	105
Age, years		3.8 ± 3.8	15 ± 2.8	41.4 ± 11.7	72 ± 7.5
Gender		Female (*n* = 1)	Female (*n* = 0)	Female (*n* = 62)	Female (*n* = 59)
		Male (*n* = 3)	Male (*n* = 2)	Male (*n* = 47)	Male (*n* = 46)
Comorbidity		2	1	66	78
ADRs		0	0	51	31
ROP	Eye Infection	1	1	41	50
	Post-surgical	-	-	31	33
	Conjunctivitis	1	-	13	7
	Chalazion	2	-	11	4
	Other	-	1	13	11
Total (ROP)		4	2	109	105
Dose	1 drop C/4 h	1	-	49	45
	1 drop C/6 h	1	1	23	22
	1 drop C/8 h	1	1	29	27
	1 drop C/12 h	1	-	7	9
	1 drop C/24 h	-	-	1	2
Total (dose)		4	2	109	105

ADR, Adverse Drug Reaction. ROP, Reason of prescription.

**Table 2 pharmacy-09-00015-t002:** ADRs of SDO vs. Dexamethasone and Ciprofloxacin.

PT.	SDO	Dexamethasone	Ciprofloxacin
Eye irritation	(*n* = 66) 30.0%	10.0% ^2^	25.0% ^2^
Vision blurred	(*n* = 3) 1.4%	9.0% ^1^	1.0% ^2^
Ocular hyperemia	(*n* = 3) 1.4%	5.0% ^1^	10.0% ^1^
Eyelid edema	(*n* = 1) 0.5%	-	1.0% ^1^
FBSE	(*n* = 1) 0.5%	-	10.0% ^1^
Dysgeusia	(*n* = 4) 1.8%	10.0% ^2^	-
Nasopharyngitis	(*n* = 1) 0.5%	4.0% ^2^	-
Eye pain	(*n* = 2) 1.0%	1.0% ^2^	-
Somnolence	(*n* = 1) 0.5%	-	-

FBSE, Foreign Body Sensation in Eyes. ADR, Adverse Drug Reaction. SDO, ciprofloxacin 0.3%/dexamethasone 0.1%. PT, Preferred Term. ID. Notes: Data from: ^1^, Micromedex [9]. ^2^, Martindale [15].

**Table 3 pharmacy-09-00015-t003:** Causality and severity of ADRs.

SOC	PT	*n*	Causality	Severity
Eye disorders	Eye irritation	66	Definite (*n* = 5), Probable (*n* = 54), Possible (*n* = 6), Doubtful (*n* = 1).	Mild (*n* = 66)
	Vision blurred	3	Probable (*n* = 1), Doubtful (*n* = 1)	Mild (*n* = 2)
			Doubtful (*n* = 1)	Moderate (*n* = 1)
	Ocular hyperemia	3	Probable (*n* = 2), Possible (*n* = 1).	Mild (*n* = 3)
	Eye pain	2	Possible (*n* = 1),	Mild (*n* = 1)
			Doubtful (*n* = 1).	Moderate (*n* = 1)
	Eyelid edema	1	Possible (*n* = 1).	Mild (*n* = 1)
	Foreign body sensation in eyes	1	Probable (*n* = 1).	Mild (*n* = 1)
Infections and infestations	Nasopharyngitis	1	Possible (*n* = 1).	Mild (*n* = 1)
Nervous system disorders	Somnolence	1	Doubtful (*n* = 1).	Mild (*n* = 1)
	Dysgeusia	4	Probable (*n* = 2), Doubtful (*n* = 2)	Mild (*n* = 1)
Total		82		

SOC, System Organ Class. PT, Preferred Term.

**Table 4 pharmacy-09-00015-t004:** Factors possibly associated with the incidence of ADRs by binomial logistic regression analysis.

Variable	B	SE	Wald	*p*	OR	95% CI. for OR	
						Lower	Upper
Gender (ref: males)							
Female	0.926	0.346	7.146	0.008 **	2.523	1.28	4.974
Comorbidities (ref: No)							
Yes	0.547	0.337	2.643	0.104	1.728	0.894	3.343
Prescription (ref: eye infection)							
Post-surgical	0.087	0.808	0.012	0.914	1.091	0.224	5.314
Conjunctivitis	−0.534	0.833	0.411	0.522	0.586	0.115	3.001
Chalazion	−0.376	0.938	0.161	0.688	0.686	0.109	4.317
Others	0.107	0.945	0.013	0.91	1.113	0.175	7.092
Age group (ref: geriatric)							
Adult	0.535	0.329	2.643	0.104	1.708	0.896	3.256
Dose (ref: 1 drop/24 h)							
1 drop/4 h	−0.074	1.31	0.003	0.955	0.929	0.071	12.104
1 drop/6 h	1.196	1.335	0.803	0.37	3.306	0.242	45.225
1 drop/8 h	0.335	1.32	0.064	0.8	1.397	0.105	18.559
1 drop/12 h	0.336	1.421	0.056	0.813	1.4	0.086	22.661

OR, odds ratio. SE, standard error. CI, confidence interval. ** *p* < 0.01.

**Table 5 pharmacy-09-00015-t005:** Bibliographic comparison of fixed combination and individual treatment by multinomial logistic regression analysis.

ADRs	B	SE	Wald	*p*	OR	95% for CI	
						Lower	Upper
Dexamethasone (ref: SDO)							
Eye irritation	−1.062	0.274	14.985	<0.001 ***	0.346	0.202	0.592
Vision blurred	1.934	0.631	9.396	0.002 **	6.915	2.008	23.813
Ocular hyperemia	1.241	0.669	3.436	0.064	3.458	0.931	12.839
Dysgeusia	1.741	0.557	9.778	0.002 **	5.705	1.915	16.994
Nasopharyngitis	2.234	1.061	4.433	0.035 *	9.336	1.167	74.696
Eye pain	0.037	1.007	0.001	0.971	1.037	0.144	7.47
Ciprofloxacin (ref: SDO)							
Eye irritation	−0.01	0.222	0.002	0.964	0.99	0.641	1.528
Vision blurred	0.172	0.826	0.044	0.835	1.188	0.235	5.998
Ocular hyperemia	2.165	0.628	11.878	0.001 *	8.712	2.544	29.839
Eyelid edema	0.865	1.231	0.494	0.482	2.376	0.213	26.535
FBSE	3.263	1.03	10.035	0.002 **	26.137	3.47	196.84

FBSE, Foreign Body Sensation in Eyes. OR, odds ratio. SE, standard error. CI, confidence interval. * *p* < 0.05, ** *p* < 0.01, *** *p* < 0.001.

## Data Availability

The data underlying this article will be shared on reasonable request to the corresponding author.

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
