# Peer review of "A Drug Safety Surveillance Study of a Ciprofloxacin/Dexamethasone Ophthalmic Fixed Combination in Peruvian Population"

_pharmacy, 2021, doi:10.3390/pharmacy9010015_

Round 1

Reviewer 1 Report

In the the Methods part, authors indicate the use of unsecured excel sheet to collect data. According to data integrity requirement, please indicate the methodology used to double check (or equivalent auto-control) the data collected  to be sure there is no transcription mistake or false data introduce in the data set.

Please indicate a reference into a clinical trial register like clinicaltrial.gov or equivalent.

In the Results part, precise siginifcation of wrong  registered number. N=17 seems very high.

In table 4 replace the title D and C by the full drug name ( the table footnote on the separate page make the table interpretation difficult).

The very small sample of some side  effect (n=1 ou 2), give give low power to

disprove the null hypothsesis; maybe for n=1 ADR the calculation of p by fisher exact test should be used carefully.

 the interesting observation difference ADR between male or female is could be correlate to a dose/body Kg  difference ? 

Author Response

Response to Reviewer 1 Comments

Point 1: In the Methods part, authors indicate the use of unsecured excel sheet to collect data. According to data integrity requirement, please indicate the methodology used to double check (or equivalent auto-control) the data collected to be sure there is no transcription mistake or false data introduce in the data set.

Response 1: The data was collected in a follow-up contact form (on paper) -added in line 89 and 90-. The information collected was recorded in a database in Excel format (password protected) by the main researcher. Subsequently, the correspondence author verified the database’s information vs paper follow-up contact form, in order to avoid errors.

Point 2: Please indicate a reference into a clinical trial register like clinicaltrial.gov or equivalent.

Response 2: In Peru and other countries the observational studies (those in which the assignment of the medical intervention is not at the discretion of the investigator) do not require registration in clinical trials; however, the evaluation of a research ethics committee in the country where the research is carried out is needed. Our study is approved by the “Comité Institucional de Bioética (CIB), Vía libre”, located in Lima, Peru (Approval Number 4206) and the pharmacovigilance plan was approved by “Dirección General de Medicamentos Insumos y Drogas (DIGEMID), Peru (code: FA-100-18).

Point 3: In the Results part, precise signification of wrong registered number. N=17 seems very high.

Response 3: Some patients did not have a personal number or cellphone, for this reason they provided a landline number that sometimes did not correspond to their home address, in most such cases it corresponded to a relative’s home. This situation created confusion when the call performed.

Point 4: In table 4 replace the title D and C by the full drug name (the table footnote on the separate page make the table interpretation difficult).

Response 4: Done. -Table 4-

Point 5: The very small sample of some side effect (n=1 or 2), give low power to disprove the null hypothesis; maybe for n=1 ADR the calculation of p by fisher exact test should be used carefully.

Response 5: Statistical analysis has been withdrawn in samples that are too small to give low power to disprove the null hypothesis. -Table 4 and added in line 180-

Point 6: The interesting observation difference ADR between male or female is could be correlate to a dose/body Kg difference? 

Response 6: The dosage and posology were analyzed through multivariate analysis -Table 3- finding no significant difference. On the other hand, this drug is applied directly to the ocular surface, and in this study, we only found local adverse reactions that are independent of mass (Kg).

Reviewer 2 Report

The question treated by the authors and the design used to collect are intersting although the bias that could be induced by the survey must be discussed deeply

The main concern about this work is the multivariate statistical method missing ! the relevance of the methodology used and the evidence strenght from the results related conclusions remain consequently very weak

Author Response

Response to Reviewer 2 Comments

Point 1: The question treated by the authors and the design used to collect are interesting although the bias that could be induced by the survey must be discussed deeply

Response 1: We used the Drug Event Monitoring design for this study (ICH topic E2E “Pharmacovigilance Planning”). https://www.ema.europa.eu/en/ich-e2e-pharmacovigilance-planning-pvp.

The survey has some limitations, -added in line 226 to 235- and also some advantages, such as individualized pharmacovigilance, continuous monitoring and access to more detailed information on adverse events from a large number of patients.

Point 2: The main concern about this work is the multivariate statistical method missing! the relevance of the methodology used and the evidence strength from the results related conclusions remain consequently very weak

Response 2: A multivariate analysis was conducted to evaluate the drug's tolerability in patients with different demographic factors such as: age group, comorbidities, gender; prescriptions and medication dosage.

The data provided by the multivariate analysis were used to strengthen the results and thus give more support to the conclusions.

-added in lines 106 to 107, 124 to 125, 159 to 164, 193 to 196 and Table 3 -

Round 2

Reviewer 2 Report

The multivariate model is still missing Which model the authors used no description is provided and where are the corresponding results 

Author Response

Point 1: The multivariate model is still missing Which model the authors used no description is provided and where are the corresponding results 

Response 1: A binary logistic regression was performed on the covariates obtained from the follow-up contact format -added in lines 123 to 126, 171 to 176, and table 4 -, and a multinomial logistic regression was performed to compare SDO vs the incidences of the literature -added in lines 127 to 129, 182 to 189 and Table 5 -.

In the case of the causality, severity, and severity of ADRs, they are considered descriptive and were not included in the multivariate analysis.
